# Clinical Distribution and Drug Resistance of *Pseudomonas aeruginosa* in Guangzhou, China from 2017 to 2021

**DOI:** 10.3390/jcm12031189

**Published:** 2023-02-02

**Authors:** Jingwen Lyu, Huimin Chen, Jinwei Bao, Suling Liu, Yiling Chen, Xuxia Cui, Caixia Guo, Bing Gu, Lu Li

**Affiliations:** 1Department of Clinical Laboratory Medicine, Guangdong Provincial People’s Hospital (Guangdong Academy of Medical Sciences), Southern Medical University, Guangzhou 510000, China; 2Guangdong Provincial Key Laboratory of Medical Molecular Diagnostics, The First Dongguan Affiliated Hospital, Guangdong Medical University, Dongguan 523808, China; 3The Fourth Affiliated Hospital of Guangzhou Medical University, Guangzhou 511316, China

**Keywords:** *Pseudomonas aeruginosa*, nosocomial infection, COVID-19, clinical distribution, drug-resistance, multidrug-resistant

## Abstract

The aim of the current study was to analyse the distribution of antimicrobial drug resistance (AMR) among *Pseudomonas aeruginosa* (*P. aeruginosa*, PA) isolates from Guangdong Provincial People’s Hospital (GDPH) from 2017 to 2021, and the impact of the COVID-19 outbreak on changes in the clinical distribution and drug resistance rate of *P. aeruginosa* to establish guidelines for empiric therapy. Electronic clinical data registry records from 2017 to 2021 were retrospectively analysed to study the AMR among *P. aeruginosa* strains from GDPH. The strains were identified by VITEK 2 Compact and MALDI-TOF MS, MIC method or Kirby–Bauer method for antibiotic susceptibility testing. The results were interpreted according to the CLSI 2020 standard, and the data were analysed using WHONET 5.6 and SPSS 23.0 software. A total of 3036 *P. aeruginosa* strains were detected in the hospital from 2017 to 2021, and they were primarily distributed in the ICU (n = 1207, 39.8%). The most frequent specimens were respiratory tract samples (59.6%). The detection rate for *P. aeruginosa* in 5 years was highest in September, and the population distribution was primarily male(68.2%). For the trend in the drug resistance rate, the 5-year drug resistance rate of imipenem (22.4%), aztreonam (21.5%) and meropenem (19.3%) remained at high levels. The resistance rate of cefepime decreased from 9.4% to 4.8%, showing a decreasing trend year by year (*p* < 0.001). The antibiotics with low resistance rates were aminoglycoside antibiotics, which were gentamicin (4.4%), tobramycin (4.3%), and amikacin (1.4%), but amikacin showed an increasing trend year by year (*p* = 0.008). Our analysis indicated that the detection rate of clinically resistant *P. aeruginosa* strains showed an upwards trend, and the number of multidrug-resistant (MDR) strains increased year by year, which will lead to stronger pathogenicity and mortality. However, after the outbreak of COVID-19 in 2020, the growth trend in the number of MDR bacteria slowed, presumably due to the strict epidemic prevention and control measures in China. This observation suggests that we should reasonably use antibiotics and treatment programs in the prevention and control of *P. aeruginosa* infection. Additionally, health prevention and control after the outbreak of the COVID-19 epidemic (such as wearing masks, washing hands with disinfectant, etc., which reduced the prevalence of drug resistance) led to a slowdown in the growth of the drug resistance rate of *P. aeruginosa* in hospitals, effectively reducing the occurrence and development of drug resistance, and saving patient’s treatment costs and time.

## 1. Introduction

*Pseudomonas aeruginosa* (*P. aeruginosa*, PA) is a conditional pathogen, that is a Gram-negative bacterium, and it is widely distributed in nature. It can parasitize human skin, the respiratory tract and the intestinal mucosa. It is characterised by a high infection rate, severe infection and rapid progress in people with low immunity. With the enhancement of drug resistance, the infection rate in hospitals is also increasing yearly [1,2]. The 30-day mortality rate of bacteraemia caused by *P. aeruginosa* is known to be as high as approximately 40% [3,4]. If not treated in time, it will lead to more serious mortality. In recent years, with the extensive and unreasonable use of antibiotics, the drug resistance of *P. aeruginosa* has gradually increased, and even multidrug-resistant bacteria, pan-drug-resistant bacteria, and all resistant bacteria have emerged, which brings severe challenges for clinical prevention and treatment and even poses the dilemma in which no drug is available. Multidrug-resistant *P. aeruginosa* (MDR-PA) involves the resistance of an isolate to at least one agent in three or more antimicrobial categories, extensively drug-resistant (XDR-PA) is an isolate defined as having nonsusceptibility to at least one agent in all but two or fewer antimicrobial categories (i.e., bacterial isolates remain susceptible to only one or two categories), and pandrug-resistant *P. aeruginosa* (PDR-PA) is an isolate defined as non-susceptibility to all agents in all antimicrobial categories [5]. In 2017, 32600 hospitalised patients in the United States were infected with MDR-PA and 2700 patients died [6], which gradually became a public health problem that cannot be ignored in hospitalised patients infected with MDR-PA [7]. *P. aeruginosa* is second only to *Escherichia coli*, *Klebsiella pneumoniae* and *Acinetobacter baumannii* in clinical infection, and it has received relatively little attention [2]. In addition, very few antibiotics can effectively treat infections caused by MDR-PA and XDR-PA [8,9]. Therefore, it is of great significance to analyse the clinical distribution and drug resistance of *P. aeruginosa* for early antibiotic treatment.

According to the surveillance report on bacterial resistance in China from 2020 (brief version), the average detection rate of carbapenem-resistant *P. aeruginosa* (CR-PA) nationwide was 18.3% [10], but China has not conducted a statistical analysis on MDR-PA and PDR-PA. Additionally, after the outbreak of COVID-19 in 2020, the Chinese epidemic prevention policy has been under strict control; hospitals have also implemented strict disinfection and isolation policies, which will also affect the drug-resistant infection and transmission of *P. aeruginosa*. Therefore, in this paper, we aim to analyse the samples of *P. aeruginosa* collected by Guangdong Provincial People’s Hospital from 2017 to 2021, and along with changes in the clinical distribution and drug resistance of *P. aeruginosa* under the COVID-19 epidemic prevention policy in 2020. The goal was to provide data support for the clinical prevention and treatment of infection and provide a theoretical basis for the study of drug resistance.

## 2. Materials and Methods

### 2.1. Source of Strain

We retrospectively reviewed the medical records of patients infected with *P. aeruginosa* who were admitted to Guangdong Provincial People’s Hospital, Guangzhou, China, from January 2017 to 2021. The study hospital is a university-affiliated and tertiary hospital with 2852 beds, with an annual discharge of 110,600 patients and an outpatient volume of approximately 4.183 million. It is one of the largest hospitals in Guangdong Province. There are 13 national key clinical specialties and 32 key clinical specialties in Guangdong Province. This hospital has three different intensive care units (ICUs) with 38 beds. All patients with *P. aeruginosa* bacteraemia who had a confirmed diagnosis based on microbiological laboratory results were included in the study over 5 years. The *P. aeruginosa* strains were isolated from clinical samples submitted by various departments of Guangdong Provincial People’s Hospital from 2017 to 2021, primarily including the Department of Critical Care Medicine, the Department of Burns, the Department of Respiratory Medicine, the Department of Paediatrics, and the Department of Haematology. Repeated test samples were eliminated, and a total of 3036 *P. aeruginosa* strains were detected. When multiple strains were isolated from the same site and from one patient within 1 month, they were designated as belonging to the same strain, with the first isolate being used as a representative sample.

### 2.2. Isolation and Identification of Strains

The isolation and culture of strain specimens are standardised by referring to the fourth edition of the National Clinical Laboratory Operating Procedures [11]. Matrix-assisted laser desorption ionization-time of flight mass spectrometry (MALDI-TOF MS) and a VITEK 2 Compact automatic bacterial identification system was used to identify the isolated and cultivated strains.

### 2.3. Antimicrobial Susceptibility Testing

The VITEK 2 Compact bacterial drug sensitivity analysis system (for matching the drug sensitivity card ASTGN) was used to detect the minimum inhibitory concentration (MIC) of drugs and perform drug sensitivity analysis. Some drug sensitivity was supplemented by the Kirby–Bauer paper diffusion method (Oxoid, Thermo Scientific™, UK). The quality control strains were *Escherichia coli* ATCC25922, *P. aeruginosa* ATCC 27853, and *Staphylococcus aureus* ATCC 29213. The interpretation of the results was performed in reference to the CLSI 2020 standard.

### 2.4. Statistical Analysis

WHONET 5.6 software and IBM *SPSS* Statistics 23.0 were used to process and statistically analyse the data related to *P. aeruginosa*. The chi-square test was used to analyse the time trend in the drug resistance rate between different groups and for the year, and results with *p*-values of less than 0.05 were considered statistically significant.

## 3. Results

### 3.1. Characteristics of P. aeruginosa Isolates from GDPH

A total of 3036 *P. aeruginosa* strains were isolated from GDPH from January 2017 to December 2021. The annual isolation of *P. aeruginosa* showed an increasing trend, but the number of isolated strains declined in 2020. The rates of MDR-PA ranged from 9.35% to 12.59% (*p* = 0.125), and the PDR-PA ranged from 0.68% to 1.17% (*p* = 0.253) over 5 years, shown in Figure 1A. During the sample separation process, the highest separation rate was found for sputum (n = 1290, 42.5%), and the rest was for wound secretion (n = 535, 17.6%), bronchoscope flushing fluid (n = 273, 9.0%), urine (n = 272, 9.0%), bronchial flushing fluid (n = 162, 5.3%), blood (n = 115, 3.8%), and others (Figure 1B).

### 3.2. Population Distribution of Strain Source

In terms of population distribution, the proportion of men infected by *P. aeruginosa* was higher than that of women, and 2104 individuals were males (68.2%), with a total male-to-female ratio of 2.14:1 (Figure 1C). The proportion of *P. aeruginosa* infections among people over 65 years old was the highest, followed by people aged 50–65, and the lowest proportion was among people aged 5–17 (Figure 1D).

### 3.3. Temporal Changes in P. aeruginosa Isolates

Through the detection rate analysis of *P. aeruginosa* during different months of the year, the largest numbers of *P. aeruginosa* strains were isolated in January 2017, October 2018, June 2019, July 2020, and September 2021, and the number of strains isolated in February was less than that in other months for the past five years (Figure 1E).

### 3.4. Department Distribution of Strains Source

According to the analysis of the detection rate for *P. aeruginosa* in each department of the hospital, the department with the highest detection rate in the distribution of departments was the Critical Care Medical Department (n = 1207, 39.8%), followed by the Respiratory Department (n = 191, 6.2%), the Burn Department (n = 182, 6.0%), the Otolaryngology Department (n = 174, 5.7%), and the General Department (n = 171, 5.6%) (Figure 2).

### 3.5. Resistance Rate of P. aeruginosa to Antibiotics

The drug sensitivity test showed that the highest *P. aeruginosa* resistance rate to imipenem was 22.4%, followed by aztreonam (21.5%), and meropenem ranked third (19.3%). The resistance rates to ciprofloxacin (13.4%), levofloxacin (11.6%), ceftazidime (10.1%) and piperacillin (10.8%) were all in the 10–15% range. The drug resistance rates to cefepime (7.5%), piperacillin/tazobactam (7.2%), gentamicin (4.4%), tobramycin (4.3%) and amikacin (1.8%) were all lower than 10% (Table 1). Further analysis of the annual change trend in the antimicrobial resistance rate and the change in single antimicrobial resistance over the past five years showed that the resistance rate of *P. aeruginosa* to imipenem, aztreonam and meropenem remained at a high level, and the difference was not statistically significant (Figure 3A). The drug resistance rate to cefepime decreased year by year (*p* = 0.001), and that of amikacin, with a lower drug resistance rate, increased year by year (*p* = 0.008). After the drug resistance rate to meropenem and piperacillin decreased in 2018 compared with that in 2017, there was an upward trend year by year over the following four years (Figure 3B). The drug resistance rate to ciprofloxacin, levofloxacin, ceftazidime, piperacillin/tazobactam, gentamicin and tobramycin was generally stable.

## 4. Discussion

*P. aeruginosa* can grow on the surface of medical equipment and form a highly resistant biofilm [12]. It can quickly mutate to obtain drug resistance to adapt to the environment and is one of the main pathogens causing nosocomial infection. Due to the extensive and irrational use of antibiotics in clinical practice, the detection rate of drug-resistant bacteria is increasing yearly [13], and its severe infection and mortality rates are also gradually increasing. A great deal of attention has been given to nosocomial infection and the generation, prevention and treatment of new drug-resistant bacteria at home and abroad [14]. A total of 3036 *P. aeruginosa* strains were isolated from January 2017 to December 2021. The annual isolation of *P. aeruginosa* showed an increasing trend, but the number of isolated strains declined in 2020, which may have been caused by the COVID-19 outbreak in 2020. The strict state of epidemic prevention and control, shutdown, control of personnel mobility, disinfection policies and other reasons may have caused this decrease in China. The number of *P. aeruginosa* isolates increased in 2021, which may be due to the relaxation of epidemic control, the resumption of work and production across the country, and the intensive movement of people. On the contrary, in contrast the MDR-PA peaked at 12.59% in 2020, and the proportion of PDR-PA peaked at 1.17% in 2021. Although the 5-year separation rates of MDR-PA and PDR-PA are statistically insignificant, the upwards trend cannot be ignored. In addition, according to the sample data from the past five years, the month with the largest number of *P. aeruginosa* isolates is not fixed, and February is the month with the smallest number of isolates, which is caused by the small total sample size in February.

Among the samples from patients infected with *P. aeruginosa* in our hospital, the samples were primarily collected from the Department of Respiratory Medicine and the intensive care unit (ICU), which was consistent with previous reports [15,16]. Among them, respiratory tract samples were the most, common because *P. aeruginosa* is an obligate aerobic bacterium, and the respiratory tract is easily infected compared with the intestinal tract and other organs. Notably, patients in the respiratory department have underlying respiratory diseases, which reduce their respiratory immunity and cause them to be easily infected with *P. aeruginosa* in the case of primary infection. However, ICU patients are in more serious condition than patients in other departments, with poor drug resistance, more antibiotic treatment and more invasive operations such as tracheal intubation. Incomplete sterilisation or improper operation of instruments will increase the risk of infection in these patients. Studies have shown that patients who are treated with a combination of two or more antimicrobial agents and stay in the ICU for more than 3 days have an increased probability of infection with carbapenem-resistant *P. aeruginosa* [17]. The wound secretions samples from our hospital were higher than those in other papers [18,19], which may be because open wounds are more susceptible to cross-infection and bacterial colonisation [20]. Therefore, medical staff should strictly implement aseptic operation in clinical work, eliminate the source of infection, prevent cross infection of medical tools, and give drug treatment for specific cases. In the treatment of large-area wounds, the use of β-lactam antibiotics with high antibacterial activity is beneficial for reducing infections by *P. aeruginosa* [20], but its disadvantage is that the frequent use of specific antibiotics or a specific type of antibiotic easily breeds drug-resistant bacteria.

The drug resistance of *P. aeruginosa* to imipenem was lower than that reported by CHINET [21] and showed a trend of decreasing first and then increasing, which was consistent with a previous report [22]. However, the drug resistance rate continued to increase from 2018 to 2020, which was inconsistent with the decreasing trend reported by CHINET [21]. The resistance of *P. aeruginosa* to imipenem is primarily caused by the production of metalloenzymes, and efflux pumps and the lack of D2 protein [23], which is speculated to be due to the widespread use of imipenem in clinical practice in recent years, and the trend of drug resistance is gradually increasing. It is suggested that a large amount of antibiotics should not be used for a long time in clinical treatment because it is easy to develop highly resistant pathogens, increasing the difficulty of treatment. The drug resistance rate of *P. aeruginosa* to meropenem increased steadily year by year, and the drug resistance rate was 20.2% in 2020, which was higher than the 19.3% reported by CHINET within the same year [21]. This upwards trend indicates that the frequency and intensity of meropenem treatment for *P. aeruginosa* infection in our hospital have increased in the past five years. The resistance rate of *P. aeruginosa* to amikacin (1.7%) is obviously cross-similar to that reported in the Korean literature (3.8–40%) [24]. It has been reported that the sensitivity of *P. aeruginosa* to polymyxin B is close to 100% [19]. Therefore, amikacin or polymyxin B is recommended for patients with severe *P. aeruginosa* infection, but the dosage should be reasonably controlled.

The type of *P. aeruginosa* resistance primarily depends on the range of its resistance to antibiotics. For MDR-PA, XDR-PA, and PDR-PA, the main clinical treatment measures are as follows. First, the treatment strategy for MDR-PA is primarily combined with antibiotics, namely, β-lactams combined with quinolone/aminoglycoside antibacterial drugs. However, specific clinical drug use should be comprehensively selected according to the drug resistance of strains, the clinical treatment effect, adverse drug reactions and differences in patients’ conditions in each country or region. In China, the treatment of MDR-PA infection primarily refers to the recommended methods in the Expert Consensus on the Diagnosis and Treatment of *P. aeruginosa* Lower Respiratory Tract Infection and the Guidelines for the Diagnosis and Treatment of HAP/VAP in Adults (2018 Edition), that is, the use of β-lactam synergistic quinolone/aminoglycoside antibacterial drugs for combined drug therapy, and the therapeutic effect of aminoglycosides combined with β-lactams was slightly stronger on *P. aeruginosa* than that of fluoroquinolones. In addition, β-lactams and fosfomycin (fosfomycin should be used one hour in advance), quinolones and aminoglycosides have good therapeutic effects on *P. aeruginosa* biofilm infection. In addition, the combination of colistin and β-lactams, ciprofloxacin, fosfomycin and other drugs is also a drug regimen for the clinical treatment of severe infections caused by Gram-negative bacteria [25,26].

However, infectious diseases caused by *P. aeruginosa* can be treated with a single drug when the drug sensitivity results are known, but it is generally not recommended to use aminoglycoside antibiotics alone for treatment. For the shock or death risk associated with *P. aeruginosa*, the combined drug treatment effect will be better [27,28,29]. According to the 2016 Clinical Practice Guidelines of the American Academy of Infectious Diseases and the American Thoracic Society, combined treatment using carbapenems and aminoglycosides may increase the risk of nephrotoxicity and ototoxicity in patients, resulting in poor treatment effects, but for patients with sepsis, the combined treatment effect is good [27]. Additionally, the HAP/VAP Management Guide (2017) shows that the combination of β-lactams and aminoglycosides can increase the therapeutic effect when treating early infections caused by MDR Gram-negative bacteria [30]. According to the research report, MDR-PA can be divided into two categories, namely, three (3-multidrug resistant Gram-negative rods, 3MRGN) and four (4MRGN) acylureidopenicillins, third or fourth generation cephalosporins, fluoroquinolones and carbapenems are resistant. Infection caused by 3MRGN-PA can be treated with ceftazidime/avibactam, while the recommendation for 4MRGN-PA is to use colistin in combination with meropenem or ceftazidime/avibactam [31]. In addition, the combination of imipenem and tobramycin can significantly enhance the bactericidal efficiency of clinically isolated MDR-PA strains [32].

Second, according to the Expert Consensus on the Diagnosis and Treatment of Lower Respiratory Tract Infection of PA in China, the therapeutic schedule for patients infected with XDR/PDR-PA is to add colistin on the basis of a combination treatment using two drugs to create a three-drug combination strategy [25]. In addition, in the “Chinese Expert Consensus on XDR Gram-Negative Bacteria Infection”, not only the treatment scheme using colistin combined with rifampicin was proposed, but also the combination treatment scheme consisting of three drugs of aztreonam combined with ceftazidime and amikacin was recommended [33]. To treat XDR/PDR-PA infection, domestic and foreign drug guidelines are basically similar, such as the Chinese Guidelines for the Diagnosis and Treatment of HAP/VAP for Adults (2018 Edition) and the Chinese Expert Consensus on the Clinical Application of Polymyxin, which put forwards the proposal that the nebulised inhalation of aminoglycosides and polymyxins based on intravenous drug use can be used to treat pneumonia caused by XDR/PDA-PA infection [26,34]. The XDR-PA cases of our hospital are shown in Appendix A. Similarly, foreign studies have reported that intravenous injection of mucin [35] is usually used for multidrug-resistant *P. aeruginosa* infection, and inhalation antibiotics can be used to treat patients with poor initial treatment effects [27]. Inhaled antibiotics (aminoglycosides or polymyxins) are only used as adjunctive therapy for infections caused by sensitive Gram-negative bacteria, but they also need to be used in combination with other antibacterial drugs [36]. In addition, it has been reported that the newly developed compound preparation of ceftolozane/tazobactam is also one of the first-choice drugs for treating XDR-PA infection [37].

Third, new antibiotics should be applied for the prevention and treatment of MDR-PA. In 2017, ceftolozane/tazobactam combined with colistin and meropenem had good efficacy in treating XDR and MDR PA infections [38,39]. In 2018, imipenem/relebactam compound preparation could reportedly enhance the bactericidal activity of imipenem and improve the bactericidal efficiency against *P. aeruginosa* [38,40]. In addition, the newly developed murepavadin antibacterial agent that was in the clinical trial stage in 2018 has a good therapeutic effect on multidrug-resistant *Klebsiella pneumoniae* and highly resistant *P. aeruginosa*; it has stronger bactericidal activity than colistin and polymyxin B, does not easily produce cross-drug resistance [41,42]. In 2020, the United States Food and Drug Administration (FDA) approved cefiderocol to treat *P. aeruginosa* producing metallic β-lactamase [43]. Taniborbactam combined with cefepime, which was tested in phase 3 clinical trials in March 2022, can be used as a broad-spectrum inhibitor of serine and metallic β-lactamase to enhance bactericidal activity against Gram-negative bacteria such as *P. aeruginosa* [44,45]. Yang, X et al. have also shown that the tobramycin-linked efflux pump inhibitor conjugate has a synergistic effect with fluoroquinolones and rifampicin and fosfomycin against MDR-PA [46].

Phage therapy was applied for the prevention and treatment of *P. aeruginosa*. Phage therapy is an investigational anti-infective treatment for refractory, MDR and/or biofilm-mediated infections. Phages are promising candidates for the treatment of recurrent and refractory infections due to their ability to target specific host bacteria and circumvent the traditional mechanisms of antibiotic resistance common in MDR/XDR-PA [47]. For biofilms formed by *P. aeruginosa*, phages can be used to eradicate *P. aeruginosa* biofilms by destroying the extracellular matrix, increasing the permeability of the inner layer of the biofilm to antibiotics, and inhibiting its formation by stopping quorum-sensing activity. In addition, the combination of phages and other compounds with antibiofilm properties, such as nanoparticles, enzymes, and natural products, may be of more interest because they invade biofilms by various mechanisms and are more effective than those used alone. However, the use of phages to destroy biofilms also has some limitations, such as a limited host range, high biofilm density, subgroup phage resistance in biofilms, and inhibition of phage infection through quorum sensing in biofilms [48]. In clinical practice, the combination of phages with antibiotics was used to treat the exacerbation of acute-on-chronic respiratory failure caused by MDR-PA, and the patient successfully underwent bilateral lung transplantation [49]. In addition, some adverse events have been associated with phage therapy, but the phage preparations administered to date are generally considered to be safe when they comply with good manufacturing practices (GMPs) or similar regulatory standards [50,51].

Last, our research has several limitations. First, due to the system-wide update of our hospital, the diagnosis and treatment data of most clinical patients from 2019 to 2020 were missing, so we did not conduct a multivariate discussion of disease status and treatment. Second, this is a single-centre study conducted in a tertiary general hospital in Guangzhou, southern China. The number of *P. aeruginosa* samples was relatively small; therefore, our results may not be extrapolated to other hospitals and regions of the country. In addition, because this is a retrospective study, we cannot rule out unmeasurable uncertainties, such as hopeless discharge in patients with cancer. However, we will continue to perform new research, and track the disease and treatment. In addition, this study lacks relevant research on drug resistance mechanisms. In future studies, we will perform a related analysis of the drug resistance mechanisms of *P. aeruginosa* to obtain a more comprehensive understanding of this issue in Guangzhou.

## 5. Conclusions

In summary, the number of clinical *P. aeruginosa* isolates in our hospital from 2017 to 2021 showed an upwards trend, and the strain samples were primarily distributed in sputum, wound secretions, etc. The infection rate of males was approximately twice that of females, and the infection rate of the elderly over 65 years old was the highest. There were different degrees of resistance to different antibiotics, and the trend in some types of resistance was slowing down, but the overall trend was increasing. This is because after the outbreak of COVID-19 in 2020, the epidemic prevention policy in China has been in a strict control state, and hospitals have also implemented strict disinfection and isolation policies, which would also have an impact on the drug-resistant infection and spread of *P. aeruginosa*. Therefore, hospitals need to do good work in bacterial resistance monitoring and infection prevention and control, and should also follow the epidemic trend in bacterial resistance, rational use of antibiotics and treatment regimens to curb the generation and spread of drug-resistant bacteria.

## Figures and Tables

**Figure 1 jcm-12-01189-f001:**
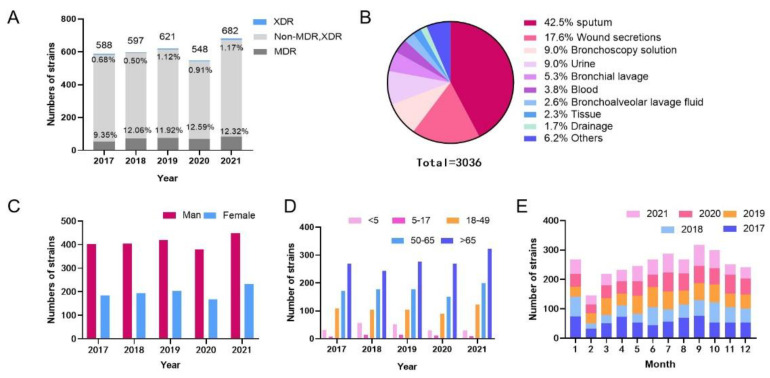
Distribution of 3036 *P. aeruginosa* strains. (**A**) Isolation of *P. aeruginosa* in 2017―2021, (**B**) Sample distribution, (**C**) Gender distribution of the population, (**D**) Age distribution of the population. (**E**) Time distribution.

**Figure 2 jcm-12-01189-f002:**
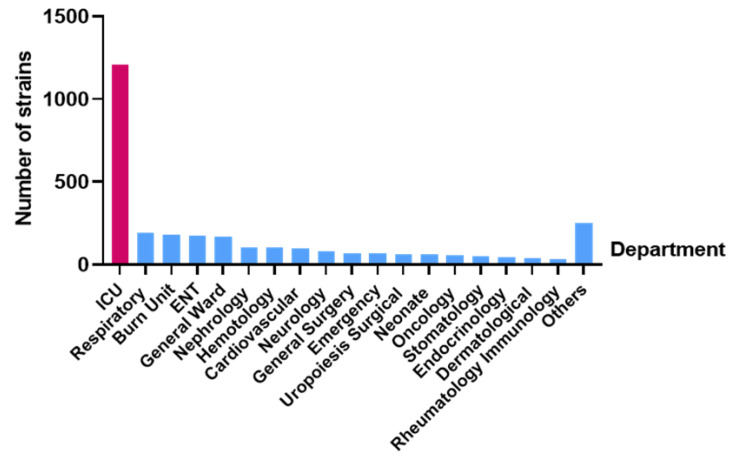
Distribution of main departments of 3036 patients infected by *P. aeruginosa*.

**Figure 3 jcm-12-01189-f003:**
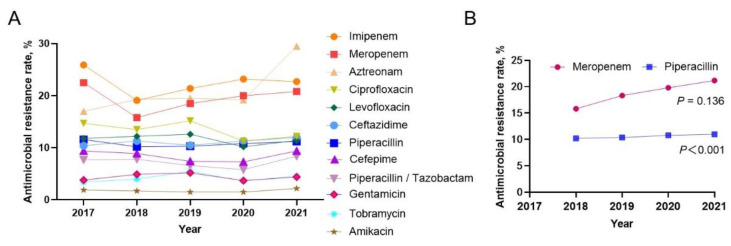
Annual antimicrobial susceptibility trends in *P. aeruginosa* from 2017 to 2021. (**A**) Changes in annual drug resistance rate of single antibacterial agents of *P. aeruginosa* from 2017 to 2021 (the note on the right for various antibacterial agents); (**B**). Annual resistance change of meropenem (*p* = 0.136) and piperacillin (*p* < 0.001) to *P. aeruginosa* from 2018 to 2021 (The change of drug resistance was statistically analysed by linear chi-square test, *p* < 0.05 indicating that the difference was statistically significant).

**Table 1 jcm-12-01189-t001:** Resistance rate of *P. aeruginosa* to antibiotics in 2017–2021 (%).

Characteristics	2017	2018	2019	2020	2021	Total	*p*
Imipenem	25.9 (151/583)	19.1 (114/596)	21.4 (133/621)	23.1 (127/548)	22.7 (155/682)	22.4 (680/3030)	0.980
Meropenem	22.5 (80/355)	15.8 (92/582)	18.5 (112/607)	20.0 (109/544)	20.8 (140/673)	19.3 (533/2761)	0.402
Aztreonam	17.0 (35/206)	19.3 (107/555)	26.7 (98/367)	22.5 (92/408)	21.2 (112/529)	21.5 (444/2065)	0.210
Ciprofloxacin	14.7 (83/565)	13.5 (80/594)	15.2 (94/620)	11.3 (62/548)	12.2 (83/681)	13.4 (402/3008)	0.120
Levofloxacin	11.4 (66/578)	12.2 (73/596)	12.6 (78/621)	10.2 (56/547)	11.4 (78/682)	11.6 (351/3024)	0.624
Ceftazidime	10.4 (61/587)	11.3 (67/594)	5.4 (32/590)	11.3 (62/547)	12.0 (82/682)	10.1 (304/3000)	0.351
Piperacillin	11.6 (41/352)	10.2 (57/560)	10.3 (62/600)	10.8 (58/538)	11.2 (74/658)	10.8 (292/2708)	0.985
Cefepime	9.4 (55/585)	8.9 (53/594)	7.4 (46/618)	7.3 (40/548)	4.8 (31/648)	7.5 (225/2993)	0.001
Piperacillin/Tazobactam	7.7 (45/582)	7.8 (46/592)	6.6 (41/617)	5.8 (32/548)	8.0 (54/676)	7.2 (218/3015)	0.668
Gentamicin	3.8 (22/579)	4.9 (29/587)	5.2 (32/620)	3.7 (20/547)	4.4 (30/680)	4.4 (133/3013)	0.974
Tobramycin	3.4 (20/586/)	4.0 (24/596)	5.5 (34/620)	3.6 (20/548)	4.6 (31/681)	4.3 (129/3031)	0.534
Amikacin	1.9 (11/585)	1.7 (10/597)	1.5 (9/619)	1.5 (8/546)	2.2 (15/680)	1.8 (53/3027)	0.730

## Data Availability

L.L. and J.L. have full access to the data and are the guarantors of the data.

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
