# Peer review of "Clinical Distribution and Drug Resistance of Pseudomonas aeruginosa in Guangzhou, China from 2017 to 2021"

_jcm, 2023, doi:10.3390/jcm12031189_

Round 1
Reviewer 1 Report
Dear authors, the current study is retrospective, carried out over a period of 5 years in a hospital in China, of Pseudomonas isolates and their antimicrobial resistance trend. The value of the study is limited, as it concerns only one hospital. There are also many translation mistakes, as well as ambiguities that must be explained :
1. there are numerous unclear or wrongly translated paragraphs: line 51-56, 87-90, 109-110, 176, 195-197, 284-285, 288-291; also terms like ceflozar or statements like β-lactamides cooperate with quinolones, patients in ICU are more serious - are wrongly translated. So extensive editing of English language and style is required.
2. please specify the size of the hospital, the number of beds, addressability
3. when defining the notion of XDR, reference is also made to Gram positives (while gram-positive cocci are only sensitive to glycopeptide antibiotics and linezolid); this is not the case; it does not represent the subject of this study.
4. there are no inclusion/exclusion criteria and the study limitation are not stated. Please include them.
5. I don't understand the meaning of subchapter 4 "Clinical treatment of drug-resistant P. aeruginosa"; seems to be more about discussions than results; and moreover, from the present study it is not specified what is the proportion of MDR/XDR strains and what resistance mechanisms have associated. Please include them and and amend this subchapter, referring to the therapeutic options used in your hospital for the treatment of patients infected with these strains.
6. could the authors give explanations for the seasonal variation of the number of isolates "the largest number of P. aeruginosa strains were isolated in September in five years, and the number of strains isolated in February, March, April and May was less than that in other months"?
Author Response
Reviewer #1:
- There are numerous unclear or wrongly translated paragraphs: line 51-56, 87-90, 109-110, 176, 195-197, 284-285, 288-291; also terms like ceflozar or statements like β-lactamides cooperate with quinolones, patients in ICU are more serious - are wrongly translated. So extensive editing of English language and style is required.
Answers: We sincerely thank the editor and all reviewers for their valuable feedback that we have used to improve the quality of our manuscript. We have revised the questions raised.
- Please specify the size of the hospital, the number of beds, addressability.
Answers: Thank the reviewer for his/her valuable feedback. The study hospital is a university-affiliated and tertiary hospital with 2852 beds, with an annual discharge of 110600 patients and an outpatient volume of about 4.183 million, it is one of the largest hospitals in Guangdong Province. There are 13 national key clinical specialties and 32 key clinical specialties in Guangdong Province. Three different intensive care units (ICUs) with 38 beds. we have added this section in line 84.
- When defining the notion of XDR, reference is also made to Gram positives (while gram-positive cocci are only sensitive to glycopeptide antibiotics and linezolid); this is not the case; it does not represent the subject of this study.
Answers: Thank the reviewer for his/her valuable feedback. The type of resistance of P. aeruginosa mainly depends on the range of its resistance to antibiotics, that is, MDR was defined as acquired non-susceptibility to at least one agent in three or more antimicrobial categories, XDR was defined as non-susceptibility to at least one agent in all but two or fewer antimicrobial categories (i.e. bacterial isolates remain susceptible to only one or two categories) and PDR was defined as non-susceptibility to all agents in all antimicrobial categories. We have modified this part in in line 57.
- There are no inclusion/exclusion criteria and the study limitation are not stated. Please include them.
Answers: Thank the reviewer for his/her valuable feedback. Repeated test samples were eliminated, and a total of 3036 P. aeruginosa strains were detected. When multiple strains were isolated from the same site and from one patient within 1 month, they were handled as belonging to the same strain, with the first isolate being used as a representative sample. We have revised the questions raised in line 96.
- I don't understand the meaning of subchapter 4 "Clinical treatment of drug-resistant P. aeruginosa"; seems to be more about discussions than results; and moreover, from the present study it is not specified what is the proportion of MDR/XDR strains and what resistance mechanisms have associated. Please include them and and amend this subchapter, referring to the therapeutic options used in your hospital for the treatment of patients infected with these strains.
Answers: We feel great thanks for your professional review work on our article. we have modified this part in paragraph in Discussion 4.
- Could the authors give explanations for the seasonal variation of the number of isolates "the largest number of P. aeruginosa strains were isolated in September in five years, and the number of strains isolated in February, March, April and May was less than that in other months"?
Answers: Thank the reviewer for his/her valuable feedback. We found that the largest number of P. aeruginosa strains were isolated in January 2017, October 2018, June 2019, July 2020, September 2021, and the number of strains isolated in February was less than that in other months for the past five years. We have revised the questions raised.

Reviewer 2 Report
Thank you for appointing me as a reviewer for this manuscript. I read it with interest. I have some comments that could be of use for the improvement of the manuscript:
· Line 58: that can be effectively treated? Probably ‘used’
· Line 63: It would be a good idea to define what MDR, XDR and PDR is (doi: 10.1111/j.1469-0691.2011.03570.x)
· Materials & Methods: Please state clearly the type of study. For example retrospective
· Line 100: the heading is difficult to understand. Please rephrase
· English revision is suggested
· Line 103: This sentence is hard to understand. Please rephrase
· Line 104-106: Please do not comment and make assumptions in the results section. Just state the data and make any comments and explanations in the discussion section
· Figure 1: subfigure 1D is not quite explained correctly in the legend. Furthermore, 1E is not explained in the legend
· One important question is whether or not the cultures shown are from individual patients. If yes, then some strains may be overrepresented, while the data on age groups and gender may also be biased
· Line 150: the correct word is gentamicin. Not gentamycin
· Figure 3: the heading of the y-axis is probably wrong. Should it be antimicrobial resistance or antimicrobial susceptibility?
· Are there any data available regarding treatment and outcomes?
· I don’t understand what section 4 is about. Generally speaking, the manuscript should be divided into an abstract, introduction, materials & methods, results, discussion, and conclusion. Section 4 reminds a review article, and its presence is ambiguous. You could add some parts from that in the discussion section but remove it. Thus, the results should be followed by the discussion section
· Please add a limitations subsection at the end of the discussion section. Some limitations include the absence of clinical data from the patients, treatment data, etc.
· Please add a conclusions section after the discussion section to summarize the main findings of this study
Author Response
Reviewer #2:
- Line 58: that can be effectively treated? Probably ‘used’.
Answers: We sincerely thank the editor and all reviewers for their valuable feedback that we have used to improve the quality of our manuscript. We have revised the questions raised in line 67.
- Line 63: It would be a good idea to define what MDR, XDR and PDR is (doi: 10.1111/j.1469-0691.2011.03570.x).
Answers: Thank the reviewer for his/her valuable feedback. The type of resistance of P. aeruginosa mainly depends on the range of its resistance to antibiotics, that is, MDR was defined as acquired non-susceptibility to at least one agent in three or more antimicrobial categories, XDR was defined as non-susceptibility to at least one agent in all but two or fewer antimicrobial categories (i.e. bacterial isolates remain susceptible to only one or two categories) and PDR was defined as non-susceptibility to all agents in all antimicrobial categories. We have revised the questions raised and cited the article in reference [5].
- Materials & Methods: Please state clearly the type of study. For example retrospective.
Answers: Thank the reviewer for his/her valuable feedback. We retrospectively reviewed the medical records of patients infected with P. aeruginosa admitted to Guangdong Provincial People's Hospital, Guangzhou, China, from January 2017 to December 2021. we have modified this part in line 84.
- Line 100: the heading is difficult to understand. Please rephrase.
Answers: Thank the reviewer for his/her valuable feedback. We have modified this heading as “Characteristics of P. aeruginosa Isolates at GDPH”.
- English revision is suggested.
Answers: We feel sorry for our carelessness. In our resubmitted manuscript, the typo is revised. Thanks for your correction.
- Line 103: This sentence is hard to understand. Please rephrase.
Answers: Thank the reviewer for his/her valuable feedback. We have revised this sentence in line 119.
- Line 104-106: Please do not comment and make assumptions in the results section. Just state the data and make any comments and explanations in the discussion section.
Answers: Thank the reviewer for his/her valuable feedback. We have revised the questions raised.
- Figure 1: subfigure 1D is not quite explained correctly in the legend. Furthermore, 1E is not explained in the legend.
Answers: Thank the reviewer for his/her valuable feedback. (D) Age distribution of the population. (E) Time distribution. We have revised the questions raised.
- One important question is whether or not the cultures shown are from individual patients. If yes, then some strains may be overrepresented, while the data on age groups and gender may also be biased.
Answers: The cultures shown are not from individual patients. When multiple strains were isolated from the same site and from one patient within 1 month, they were handled as belonging to the same strain, with the first isolate being used as a representative sample.
- Line 150: the correct word is gentamicin. Not gentamycin
Answers: We were really sorry for our careless mistakes. Thank you for your reminder. We have revised the questions raised.
- Figure 3: the heading of the y-axis is probably wrong. Should it be antimicrobial resistance or antimicrobial susceptibility?
Answers: We were really sorry for our careless mistakes. Thank you for your reminder. Our response is given in changes to the manuscript are given in the chart.
- Are there any data available regarding treatment and outcomes?
Answers: We have supplemented the data on patient treatment, please refer to the supplementary table 1 in line 289.
- I don’t understand what section 4 is about. Generally speaking, the manuscript should be divided into an abstract, introduction, materials & methods, results, discussion, and conclusion. Section 4 reminds a review article, and its presence is ambiguous. You could add some parts from that in the discussion section but remove it. Thus, the results should be followed by the discussion section
Answers: We feel great thanks for your professional review work on our article. We have revised the questions raised.
- Please add a limitations subsection at the end of the discussion section. Some limitations include the absence of clinical data from the patients, treatment data, etc.
Answers: We feel great thanks for your professional review work on our article. We have revised the questions raised and added this section in line 335.
- Please add a conclusions section after the discussion section to summarize the main findings of this study.
Answers: We feel great thanks for your professional review work on our article. We have revised the questions raised.

Round 2
Reviewer 1 Report
Dear authors, thank you for your comments and answers.
The revised paper could be accepted now
Reviewer 2 Report
The manuscript has been improved during the revision process.